# Candidate genes under selection in song sparrows co-vary with climate and body mass in support of Bergmann's Rule

Katherine Carbeck [1] ✉, Peter Arcese [1], Irby Lovette [2,3], Christin Pruett[4], Kevin Winker[5] & Jennifer Walsh [2]

Ecogeographic rules denote spatial patterns in phenotype and environment that may reflect local adaptation as well as a species' capacity to adapt to change. To identify genes underlying Bergmann's Rule, which posits that spatial correlations of body mass and temperature reflect natural selection and local adaptation in endotherms, we compare 79 genomes from nine song sparrow (*Melospiza melodia*) subspecies that vary ~300% in body mass (17 – 50 g). Comparing large- and smaller-bodied subspecies revealed 9 candidate genes in three genomic regions associated with body mass. Further comparisons to the five smallest subspecies endemic to California revealed eight SNPs within four of the candidate genes (*GARNL3*, *RALGPS1*, *ANGPTL2*, and *COL15A1*) associated with body mass and varying as predicted by Bergmann's Rule. Our results support the hypothesis that co-variation in environment, body mass and genotype reflect the influence of natural selection on local adaptation and a capacity for contemporary evolution in this diverse species.

Explicating correlations among phenotypes and environmental variables is a historic focus of theoretical and empirical research on natural selection, local adaptation, and speciation[1–4]. More recently, questions about local adaptation and the adaptive capacity of species to persist given environmental change have reinvigorated efforts to understand the micro-evolutionary processes involved. For example, correlations between ambient temperature, phenotype, and life-history traits affecting fitness and subject to natural selection are widely reported in vertebrates[2,5–8] and summarized as ecogeographical rules[9]. Of particular relevance to climate adaptation, Bergmann's rule posits an inverse correlation between ambient temperature and body size in endotherms[10] and has garnered substantial empirical support (birds:[11–13]; mammals:[14,15]). If such patterns that follow Bergman's rule reflect local adaptation in traits affecting fitness, such as thermal tolerance, we expect genes underlying such trait variation to show signs of historical or ongoing selection and to vary predictably with phenotype.

Species with large ranges that encompass steep environmental gradients are excellent candidates in which to test for a genomic basis of local adaptation[16]. We focused on song sparrows (*Melospiza melodia*), which are among the world's most variable species[17,18] as indexed by the number of recognized subspecies (25), and which accordingly exhibit a stunning range of co-variation in phenotype, life history, and environment (e.g.,[11,19,20]). Song sparrows inhabit highly heterogeneous environments, spanning about 36 degrees of latitude (26 °N to 62 °N) and 15 °C in mean annual temperatures (ca. 1 °C to 16 °C). Because many phenotypic traits of song sparrows are known to have an additive genetic basis, respond to selection, and affect individual fitness (e.g.,[21–23]), we hypothesized that the phenotypic clines in body mass exemplified in song sparrows at least partly reflect local adaptation to environment. If so, we further suggest that identifying the genomic mechanisms underlying such patterns can inform us about the adaptive capacity of other species displaying local adaptation to heterogeneous environments.

[1]Department of Forest and Conservation Sciences, University of British Columbia, Vancouver, BC T6T 1Z4, Canada. [2]Fuller Evolutionary Biology Program, Cornell Lab of Ornithology, Cornell University, Ithaca, NY 14850, USA. [3]Department of Ecology and Evolutionary Biology, Cornell University, Ithaca, NY 14850, USA. [4]Department of Biology, Ouachita Baptist University, Arkadelphia, AR 71998, USA. [5]University of Alaska Museum, University of Alaska Fairbanks, Fairbanks, AK 99775, USA. ✉e-mail: katherinecarbeck@gmail.com

Here, we test this idea by comparing 40 whole genomes of two large- (*M. m. maxima*, *M. m. sanaka*; mean mass [range]: 45.9 [41.9–50.0 g]) and two smaller-bodied subspecies (*M. m. merrilli*, *M. m. rufina*; mean mass [range]: 26.7 [21.9–30.9 g]) that breed in northern British Columbia and Alaska in order to evaluate the genomic landscape of divergence and identify candidate genes associated with body mass. *Maxima* and *sanaka* reside year-round from the Alaskan Peninsula to the Aleutian Islands, experience cold winters and are nearly twice the mass of *rufina* and *merrilli*, which experience warmer winters[24] (Fig. 1). With candidate genes for body mass identified, we then use those candidates to predict genotype in 39 birds that represent the five smallest subspecies of song sparrows, all endemic to California and residing in or adjacent to San Francisco Bay (mean mass [range]: 19.2 [16.9–23.3 g]). By doing so, we demonstrate the wider utility of identifying candidate genes that are targets of selection and which contribute to local adaptation in phenotypically variable species that occupy a diverse range of environmental conditions and geographic scales.

## Results

### Phenotypic correlations and genomic divergence

Body mass (g) and mean winter and summer temperatures (°C) were strongly negatively related in the subspecies we studied (Fig. 1), consistent with Bergmann's rule. At a genome-wide scale, we observed marked differentiation and unambiguous clustering between all four northern subspecies, clearly distinguished as large- (*maxima* and *sanaka*) and smaller-bodied (*merrilli* and *rufina*) subspecies (11,223,039 SNPs; PC1: 19.93% and PC2: 6.76% of genomic variation; Supplementary Fig. 1). This pattern holds when the southern small-bodied subspecies from San Francisco Bay are included, with a PCA delineating body size on PC axis 1, which accounted for 12.48% of genomic variation (PC2: 3.48%; Fig. 1b). Population structure at $K = 2$ had the lowest cross-validation error, corresponding to grouping of the large- versus smaller-bodied subspecies; but structuring was apparent at $K = 4$ (ADMIXTURE; Supplementary Fig. 2). Genome-wide estimates of $F_{ST}$ ($\pm$ SD) indicated substantial divergence, with pairwise genome-wide values from $0.035 \pm 0.026$ in the control comparisons (*merrilli-rufina*) to $0.247 \pm 0.112$ (*maxima-merrilli*; Supplementary Table 1). Similarly, the genome-wide absolute genetic divergence (*Dxy*) was lowest between the control comparisons (mean $\pm$ SD; *maxima-sanaka*: $0.292 \pm 0.087$) and highest between *maxima-merrilli* ($0.425 \pm 0.046$). Overall, the strongest signals of genetic differentiation were best explained by body size.

### Identification of candidate genes

We identified a total of 25 elevated windows of $F_{ST}$ that were shared by at least two pairwise comparisons of large- and smaller-bodied subspecies (i.e., windows exhibiting $F_{ST}$ estimates in the 99.9th percentile of the genome-wide mean; Fig. 2). Several of these shared elevated 50 kb windows contained annotated genes (13 genes: *sanaka* and *merrilli*; 10: *sanaka* and *rufina*; 18: *maxima* and *rufina*; 19: *maxima* and *merrilli*). Overall, the mean $F_{ST}$ values in the 50 kb outlier windows were elevated for each of the four pairwise comparisons (Mean $F_{ST}$: *sanaka-merrilli* = 0.671; *sanaka-rufina* = 0.625; *maxima-rufina* = 0.645; *maxima-merrilli* = 0.828), relative to low genome-wide $F_{ST}$ averages (*sanaka-merrilli* = 0.159; *sanaka-rufina* = 0.176; *maxima-rufina* = 0.205; *maxima-merrilli* = 0.247; Fig. 2).

To specifically identify putative genes under selection for body size, we focused on genes that were shared between all four large- vs smaller-bodied comparisons (i.e., "candidate genes"). Six genes were shared by two subspecies comparisons (*AIDA*, *TPPP*, *USP16*, *LTN1*, *C3ORF52*, and unknown gene *ENSTGUP00000002762*), and two shared by three comparisons (*TRIP13* and *BRD9*; Supplementary Table 2). Notably, across the shared 50 kb windows, 9 candidate genes were identified in all large- and smaller- bodied pairwise comparisons (*FBXW2*, *GARNL3*, *RALGPS1*, *ZBTB34*, *ANGPTL2*, *ZBTB43*, *COL15A1*,

*TGFBR1*, *TAF1A*), with six candidate genes occurring on the same scaffold (contig 391/chr 17; Supplementary Table 2). None of these candidate genes were shared in control comparisons between the two larger (*maxima* and *sanaka*) and smaller (*merrilli* and *rufina*) subspecies pairs, as expected if these differentiated regions arose via divergent selection on body size (Fig. 3; Supplementary Table 2).

### Signatures of selection

To search for evidence of local adaptation in body size, we calculated the composite likelihood ratio (CLR) test statistic to test for the presence of selective sweeps on the 3 contigs containing the 9 candidate genes (Fig. 3). Selective sweeps were evident on all contigs (>99th percentile of contig-wide mean; Fig. 3). We also measured nucleotide diversity ($\pi$) and Tajima's *D* across 50 kb windows in all four northern subspecies to characterize genetic diversity and detect evidence of selection. Commensurate with our hypothesis of natural selection, nucleotide diversity was reduced in the regions of the 9 candidate genes shared among all four comparisons (*maxima* = 0.0001, *sanaka* = 0.0002, *merrilli* = 0.0002, *rufina* = 0.0005), whereas the genome-wide average was an order of magnitude higher (*maxima* = 0.0016, *sanaka* = 0.0022, *merrilli* = 0.0032, *rufina* = 0.0026) (Supplementary Figs. 3–5). Tajima's *D* was also reduced in candidate regions, especially in *maxima* (−0.332), *sanaka* (0.229), and *merrilli* (−0.789), as compared to genome-wide means (0.851, 0.892, and 0.889, respectively), but not in *rufina* ($D_{Taj\ candidate} = 1.038$; $D_{Taj\ mean} = 1.219$). *Dxy* estimated in candidate regions was also elevated in size-related pairwise comparisons (*maxima-merrilli* = 0.899, *maxima-rufina* = 0.710, *sanaka-merrilli* = 0.844, *sanaka-rufina* = 0.771) compared to controls of within-size comparisons (*maxima-sanaka* = 0.032, *merrilli-rufina* = 0.233; Supplementary Fig. 6). Taken together, these results provide evidence for the presence of selective sweeps and imply that the strength and direction of selection may vary among subspecies.

### Validation of candidate genes

Given the identification of 9 genes associated with body size and shared by the 4 northern subspecies, we next tested if these candidate genes could be used to predict allele frequency commensurate with the smaller-bodied phenotype in the 5 smallest subspecies of song sparrows endemic to California (mean mass [range]: 19.2 [16.9–23.3 g]), for which we had prior genomic data[25]. Allele frequency of the non-reference allele was assessed across the 467 SNPs located within the 9 candidate genes noted above revealed 8 SNPs that were highly correlated with mass ($r > 0.90$; $p < 0.001$; Fig. 4). Four of these SNPs were located within *RALGPS1* (one of which is also in *ANGTPL2*), 1 SNP in *GARNL3*, and 3 SNPs in *COL15A1* (Fig. 4). One SNP on *RALGPS1* (position 57980) falls within the coding region of the gene, while the remaining SNPs are located within non-coding regions of genes. Notably, non-reference allele frequencies of the 8 candidate SNPs were also highly negatively correlated with average winter and summer temperature (range $r = -0.79$ to −0.85; $p < 0.05$; Fig. 4).

We observed consistent positive regressions of mass on non-reference allele frequency at each of the 8 focal SNPs (range: $F(1,7) = 31.96–142.66$; range $R^2$: 0.820–0.953; $p < 0.001$; Supplementary Table 3). In contrast, regressing temperature on allele frequency revealed negative relationships at each of the focal SNPs (range: $F(1,7) = 11.38–18.15$; range $R^2$: 0.619–0.722; $p < 0.05$; Supplementary Table 4). We used a partial Mantel test to further explore whether these relationships reflect a history of selection versus neutral processes using the 3 contigs containing the 9 candidate genes. Pairwise genetic distance and phenotype were significantly correlated across all 9 subspecies after controlling for geographic distance (partial Mantel test: $r = 0.183$; $p = 0.004$).

Per-site $F_{ST}$ estimates for the 8 focal SNPs noted ranged from 0.63 to 0.96 among all large- and smaller-bodied subspecies (mean: 0.713–0.763, while control comparisons ranged from 0 to 0.272 (mean:

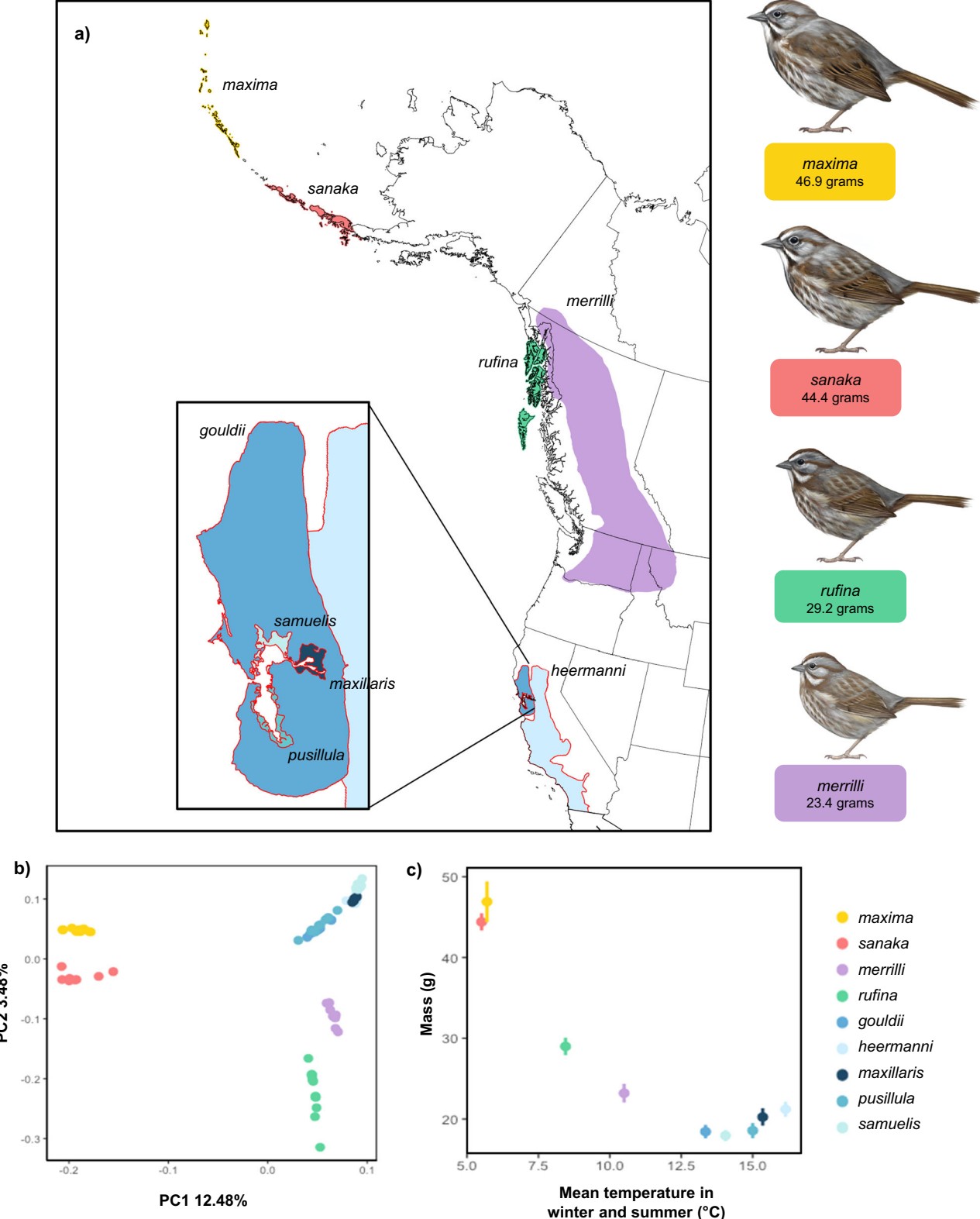

**Fig. 1 | Song sparrow subspecies distribution, genetic divergence, and body size variation. a** Range map of song sparrow subspecies (digitized and georeferenced from ref. 16) with the focal subspecies labeled therein. Inset map shows the ranges of small-bodied San Francisco Bay subspecies (outlined in red) that were used in analyses to validate candidate genes. **b** Genome-wide patterns of divergence between large- (*maxima*: yellow; *sanaka*: pink), smaller- (*rufina*: green; *merrilli*: purple), and small-bodied (*gouldii, heermanni, maxillaris, pusillula,* and *samuelis*: shades of blue) subspecies of song sparrows based on 11,223,039 SNPs. Principal

component analysis plots show clear splits between northern subspecies, with California subspecies clustering together. PC1 and PC2 explained 12.48% and 3.48% of the total variation, respectively. **c** Mean winter (Dec, Jan, Feb) and summer (Jun, Jul, Aug) temperature (°C) and mean (± SD) song sparrow body mass (g) by subspecies (n: *maxima* = 12; *sanaka* = 8; *rufina* = 12; *merrilli* = 8; *gouldii* = 10; *heermanni* = 8; *maxillaris* = 6; *pusillula* = 9; *samuelis* = 6; see methods). Illustrations depicting the average body size of the northern subspecies (illustrations by Jillian Ditner 2022).

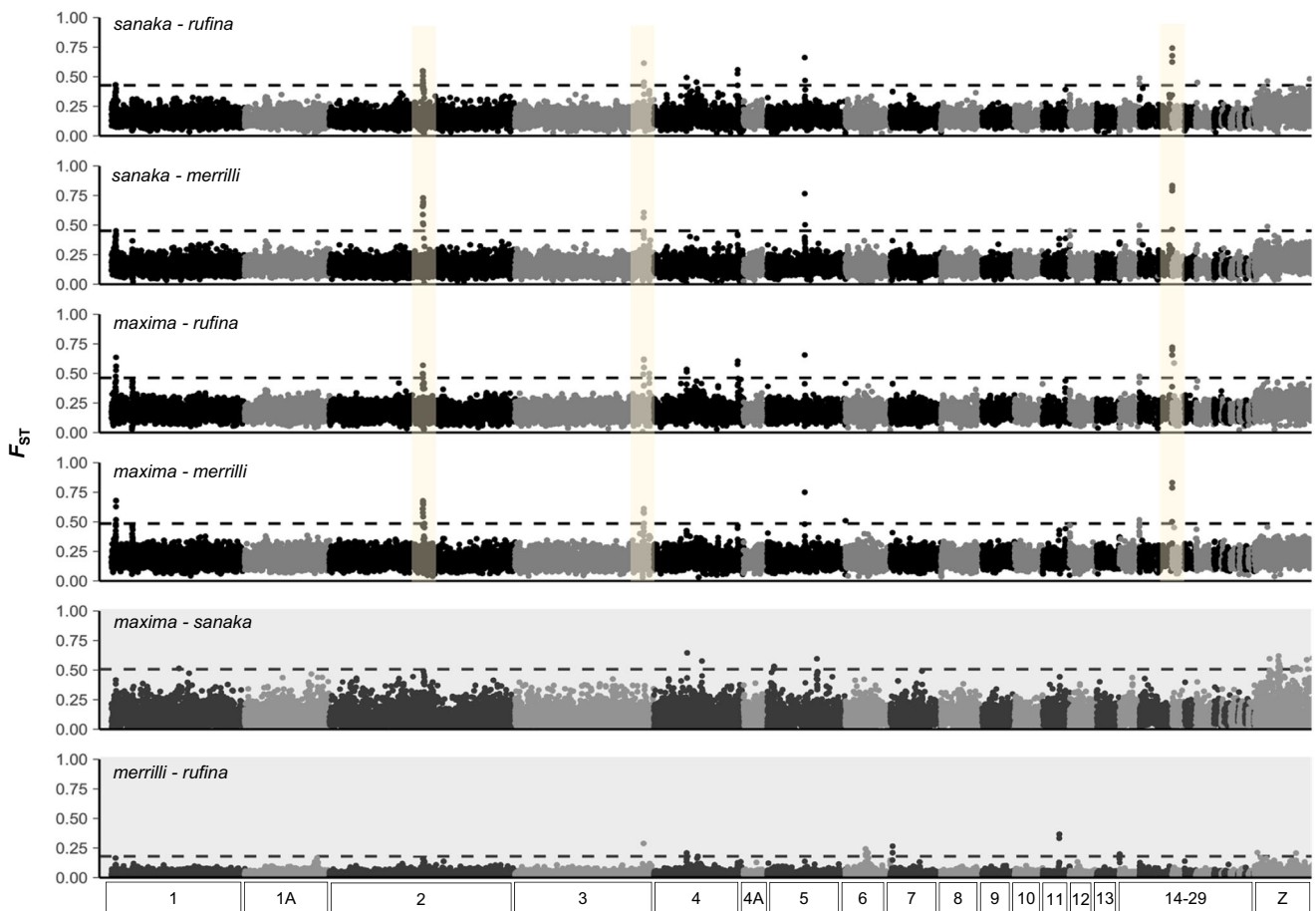

**Fig. 2 | Genome-wide differentiation between northern subspecies.** Genome-wide distribution of $F_{ST}$ for pairwise comparisons between large- and smaller-bodied northern subspecies (white background) and control comparisons (gray background). Manhattan plot show genome-wide differentiation in 50-kb windows. The dashed line indicates the 99.9th percentile of genome-wide mean. The yellow highlighted regions indicate a subset of divergent windows on contigs 3361 (chr 2), 1534 (chr 3), and 391 (chr 17) held in common between all pairs. Chromosomes were identified by their position on the zebra finch genome.

0.003–0.081; Supplementary Table 5). Across the 9 candidate genes, the percentage of fixed or nearly fixed SNPs between the large- and smaller-bodied subspecies ranged from 3.0 to 33.8%, whereas 0% of control comparisons contained fixed SNPs (Supplementary Table 5).

A phylogenetic tree reconstructed from individual genotypes of the 8 focal SNPs and 1–3 additional SNPs immediately up and downstream (54 total SNPs; Supplementary Fig. 7; see Methods) was consistent with the strong correlation between body size and non-reference allele frequency across subspecies. Across the nine subspecies, three clusters formed largely reflecting body size and geography, comprising two subspecies of intermediate mass (*rufina, merrilli*), two large Aleutian endemics (*maxima, sanaka*), and five small California endemics (*gouldii, heermanni, maxillaris, pusillula, samuelis*). Whilst these findings suggest compelling support for our hypothesis that historical and/or ongoing natural selection has contributed to genetic differentiation of body size between subspecies, lower coverage for the San Francisco Bay populations may limit our ability to fully resolve heterozygotes and should be validated using higher coverage, phased sequence data.

## Discussion

We characterized 9 candidate genes associated with body mass in 4 subspecies of large- and smaller-bodied song sparrows that breed at the northern extent of the species' range, and then used those candidates to predict genotype in 39 individuals from 5 small-bodied subspecies endemic to California that are ~50–70% smaller than subspecies that reside year-round in Alaska. Bergmann[10] explained the

tendency for endotherms to be larger in colder environments as an adaptation to minimize heat loss, though mechanisms remain uncertain[13,26]. Our observations of fixed and shared genomic differences, their co-variation with climate and body mass, and evident signatures of selection support Bergmann's Rule as influential in song sparrows reflect a capacity for and history of local adaptation to environment in this species. Moreover, because many other traits exhibit substantial additive genetic variation[23,27], respond to selection[21,28], co-vary with environment[11,25], and affect individual fitness in this species[23], we interpret these cumulative results as suggesting that song sparrows exhibit substantial capacity for contemporary evolution via context-dependent selection on individual phenotype and life history[20,29].

We observed multiple regions of elevated divergence in the genomes of large- vs smaller-bodied song sparrows versus a relatively homogenous background (Fig. 2; Supplementary Fig. 6), extending prior demonstrations of local adaptation at micro- to macrogeographic scales[20,25,30]. Consistent clustering of genetic groups in the large- and smaller-bodied populations was evident in comparisons based on mass and subspecies identity (Fig. 1; Supplementary Fig. 1) but absent in shared regions of divergence in control comparisons among subspecies of similar size (Fig. 2; Supplementary Fig. 6). These contrasts also support our hypothesis that local adaptation in song sparrow is underlain, as least in part, by genes that play a causal role in the patterns Bergmann sought to explain.

Having identified candidate genes capable of differentiating large- and smaller-bodied subspecies in Alaska, we then validated these

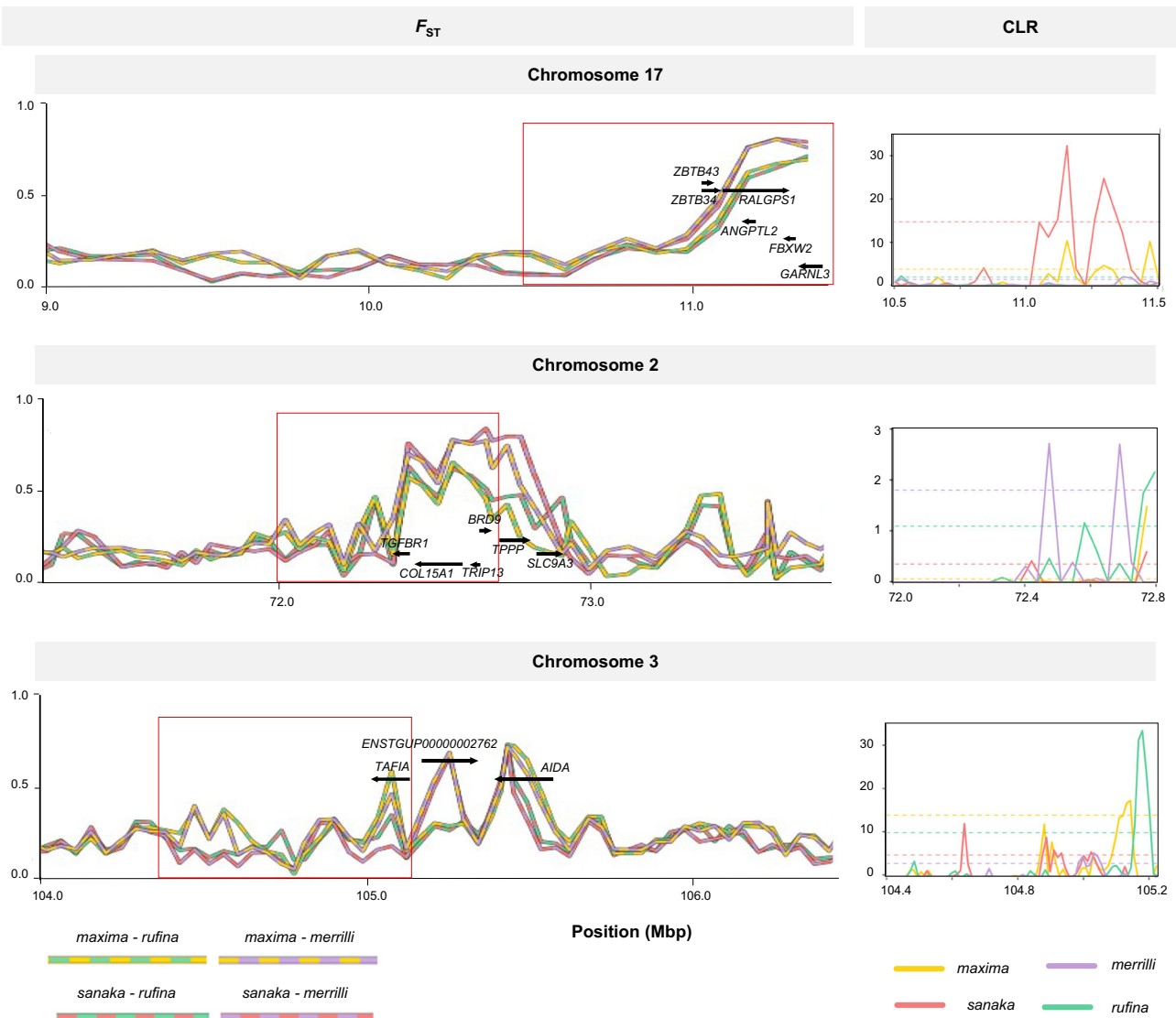

**Fig. 3 | Genetic differentiation and selective sweeps within candidate genes.** Pairwise $F_{ST}$ comparisons and composite likelihood ratio (CLR) test statistic values within regions of the 9 candidate genes. The distribution of $F_{ST}$ in 25-kb windows and genes within the divergent windows on Chromosome 17 (contig 391), Chromosome 2 (contig 3361), and Chromosome 3 (contig 1534) for pairwise comparisons between large- (*maxima*: yellow; *sanaka*: pink), smaller-bodied (*rufina*: green; *merrilli*: purple) northern subspecies. CLR values and 99th percentile of the contig mean (dashed lines) are shown within a subset of the region within the red box are suggestive of selective sweeps.

patterns by predicting the genotypes of 39 birds from the five smallest subspecies endemic to California. Of the 9 candidate genes in regions shared by all large- and smaller-bodied pairwise comparisons, four contained SNPs ($r > 0.90$) that predicted allele frequency of small-bodied subspecies in California (*RALGPS1*, *COL15A1*, *GARNL3*, and *ANGTPL2*; Fig. 4), and included genes linked to body mass index (BMI), height, and fat distribution in humans (Supplementary Table 2; see methods). *RALGPS1* (mean $F_{ST}$: 0.797, range: 0.071–0.974) and *COL15A1* (mean $F_{ST}$: 0.514, range: 0.010–0.945) similarly associate with percent and distribution of body fat, size and BMI in humans[31,32]. *ANGPTL2* contained several highly differentiated SNPs correlated with body mass and temperature (mean $F_{ST}$: 0.917, range: 0.845–0.959). *ANGPTL2* codes for angiopoietin-like protein 2 which has important roles in lipid, glucose and energy metabolism across taxa, and has also been found to be associated with human height and obesity[33,34]. In contrast, *GARNL3* (mean $F_{ST}$: 0.791, range: 0.055–0.973) was informative in differentiating subspecies by body size and predicting the genotype of smaller-bodied song sparrows, but it has not yet been linked to phenotype in other species. Notably, there were high proportions of fixed or nearly fixed SNPs ($F_{ST} > 0.95$) within *RALGPS1*, *GARNL3*, and *ANGTPL2* (18/151; 7/59; 6/14 SNPs, respectively). These results support further our supposition that some or all of the genes we identified are directly or indirectly associated with thermoregulatory capacity and influential in their contributions to the roughly 300% increase in the body mass of song sparrows observed from California to Alaska.

Although five of the candidate genes did not include SNPs highly correlated with body mass in small-bodied subspecies endemic to California, four of these genes have previously been associated with BMI, body fat mass, height, or Marfan syndrome in humans (e.g., *FBXW2*, *ZBTB43*, *TGFBR1*, *TAF1A*;[35–40]; Supplementary Table 2), whereas *ZBTB34* appears to reflect a novel association to mass. Because complex phenotypes often arise via many genes of small effect[41], it is likely that such smaller effects would be challenging to discern given our statistical approach, stringent identification of candidates, and modest sample sizes.

Despite strong associations between body mass and the candidate genes noted above, genetic drift, variation in recombination rate and

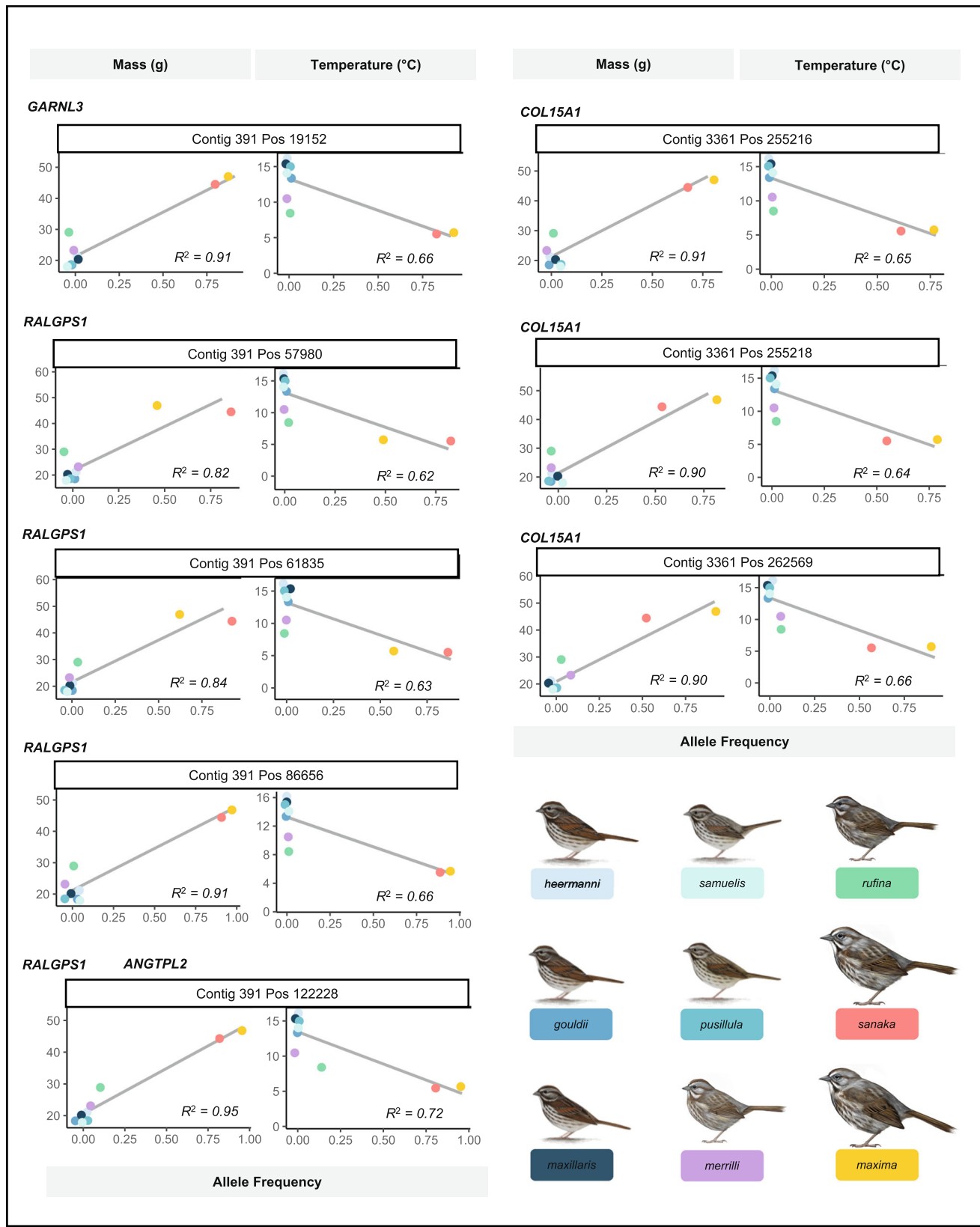

genetic architecture, and selection on correlated traits are also expected to influence population-level variation in the phenotype of song sparrows across their North American range[42]. Thus, whilst genome scans clearly offer valuable insights into the origins of adaptive evolution[43–45], more detailed genomic and field studies are needed to validate gene expression, the effects of natural selection on the adaptive capacity of populations, phenotype, and fitness, and to understand the consequences of immigration on genetic architecture. Methods such as QTL mapping of hybrid individuals in populations with precise pedigrees will be needed to elucidate the additional effects of chromosomal rearrangements, insertions, deletions, or duplications, on phenotype, plasticity, and the process of speciation[23,29]. Similarly, a more precise

**Fig. 4 | Relationship between non-reference allele frequency, body mass, and temperature for highly divergent SNPs.** Allele frequency of the non-reference allele vs mass (g) and temperature (°C) for eight highly divergent SNPs within four of the 9 candidate genes. The SNPs of interest are located within $F_{ST}$ peaks on contigs 391 (chr 17) and 3361 (chr 2) and located within five of the eight candidate genes (*GARNL3*, *RALGPS1*, *ANGTPL2*, and *COL15A1*) of *maxima* and *sanaka* (large-bodied; yellow and pink, respectively), *rufina* and *merrilli* (smaller-bodied; purple and green, respectively), and *M. m. gouldii*, *M. m. heermanni*, *M. m. maxillaris*, *M. m. pusillula*, and *M. m. samuelis* (small-bodied; shades of blue) (illustrations by Jillian Ditner 2022). Non-reference allele frequency is positively correlated with body mass (g) and negatively correlated with average winter and summer temperature (°C). Points are color coded by subspecies identity; solid black line represents the line of best fit and shaded area represents standard error, with coefficient of determination values ($R^2$) reported in lower right of plots.

understanding of the influence of plasticity versus direct or indirect selection on traits underlying Bergmann's or other ecogeographic rules is needed to estimate the capacity of populations to maintain fitness given environmental change[13,26,46,47].

Song sparrows exhibit a stunning range of variation in external phenotype and life history that correlates closely with local environmental conditions[11,18–20,25]. Because many phenotypic traits that affect individual fitness have an additive genetic basis in this species[21,23,28], it is plausible that spatial and temporal variation in natural selection have contributed substantially to local adaptation and divergence across its range[11,18–20,25]. Our results reveal strong evidence of natural selection acting on genes which appear to contribute to phenotypic variation in body mass among nine subspecies distributed over a dramatic environmental cline, and augment our prior work demonstrating local adaptation in the genomes of individuals across macro- to microgeographic scales[25,48]. Furthermore, our novel approach to validating gene function strongly suggests that the same genes have experienced historic and/or ongoing selection in multiple populations across the species' range. Our findings thus support and extend our prior results at local to landscape scales[23,25,30], implying a substantial capacity for eco-evolutionary adaptation to environment change in the song sparrow, and reinforcing its value as a model species for understanding the genomic underpinnings of adaptive evolution.

## Methods
### Study system and sampling
To identify candidate genes underlying Bergmann's rule, we intentionally sampled large- and smaller-bodied song sparrows from the northern extent of the range where these populations in close geographic proximity vary substantially in mass. In total, we sampled 40 male song sparrows representing two large-bodied subspecies, *M. m. maxima* (n = 12; mean ± SD = 46.9 ± 2.5 g) and *M. m. sanaka* (n = 8; 44.4 ± 1.1 g), and two smaller-bodied subspecies, *M. m. merrilli* (n = 8; 23.2 ± 1.2 g) and *M. m. rufina* (n = 12; 29.0 ± 1.1 g). Tissue samples for these individuals were provided by the University of Alaska Museum. All birds were collected between 1997 and 2000 (Supplementary Data 1). We then used these candidate genes to predict the genotypes of small-bodied subspecies from California: *M. m. gouldii* (n = 10; mean ± SD = 18.44 ± 0.84 g); *M. m. heermanni* (n = 8; 21.22 ± 0.95 g); *M. m. samuelis* (n = 6; 17.95 ± 0.70 g); *M. m. pusillula* (n = 9; 18.57 ± 0.95 g); and *M. m. maxillaris* (n = 6; 20.25 ± 1.08 g). Because extreme cold and heat are both expected to influence body size we used mean winter (Dec, Jan, Feb) and summer (Jun, Jul, Aug) temperature data, which were acquired from ClimateNA v.6.00[49] for each of the sample locations to visualize the relationship between body mass and temperature (Fig. 1). All subspecies are year-round residents across their range except *rufina* and *merrilli*, which are considered to be partial migrants. At our sampling locations, however, *rufina* are known residents and *merrilli* are known migrants, therefore we estimated the mean winter temperature for *merrilli* based on Patten and Pruett[18].

### Whole-genome sequencing and variant discovery
Genomic DNA was extracted using the DNeasy Blood and Tissue Kit (Qiagen, CA, USA). DNA concentrations were quantified using the Qubit BR dsDNA Assay Kit (Life Technologies). Using 150 ng of DNA from each sample, we prepared individually barcoded libraries with a 300 bp insert size following the protocol for the NEBNext Ultra II FS

DNA Library Prep Kit (Illumina, CA, USA). Libraries for *M. m. sanaka* and *M. m. merrilli* were sequenced on a single Illumina NextSeq lane at the Cornell Institute for Biotechnology core facility, while those of *M. m. maxima* and *M. m. rufina* were sequenced on a single Illumina NovaSeq lane at Novagene (The University of California at Davis campus). Whole-genome sequencing yielded a mean of 48,428,940 reads per individual in our British Columbia and Alaska populations.

We assessed library quality using FastQC v.0.11.8 (http://www.bioinformatics.babraham.ac.uk/projects/fastqc). We used Adapter-Removal v.2.1.1 for sequence trimming, adapter removal, and quality filtering (requiring a minimum Phred quality score of 30), and we merged overlapping paired-end reads. We aligned filtered reads to the song sparrow reference genome based on a small-bodied subspecies (*M. m. morphna*) from Southern British Columbia[50] using the default settings in Bowtie2 v.2.4.2[51] and obtained alignment statistics from Qualimap v.2.2.1[52]. Mean alignment rates for the four subspecies comparisons were 97.98%. Mean coverage and mapping quality were 7X (range: 4X–16X), and 18.62 (Table S3). We used SAMtools v.1.9[53] to convert all resulting SAM files to BAM files and to sort and index files. We used Picard Tools v.2.8.2 (https://broadinstitute.github.io/picard/) to add index groups and mark duplicates. We used mpileup module in Bcftools v.1.12 for SNP variant discovery and genotyping for all song sparrows and filtered out variants that were not biallelic, had minor allele frequencies less than 5%, mean coverage less than 2X or more than 50X, and more than 20% missing data. This resulted in a total of 13,089,663 SNPs across the four subspecies. The mean missing data across individuals was 16.9% (Supplementary Data 1). We identified three samples that exhibited 50% or greater relatedness to another individual. Because three (*rufina*, *sanaka*, and *maxima*) of our subspecies were sampled from islands and are likely to have higher levels of inbreeding[54], this signal is likely biologically relevant. To assess whether retaining related individuals had any impact on population structure, exploratory PCA plots were run both with and without these individuals. This comparison produced similar results (Fig. 1, Supplementary Fig. 1); therefore, all downstream analyses were conducted with the full dataset, including related individuals.

### Population genomic analyses
We visualized genetic clustering in the SNP dataset by performing a PCA using snpgdsPCA function in the SNPRelate package[55] in R v.4.0.5[56]. To quantify the level of genome-wide differentiation between subspecies, we calculated $F_{ST}$[57] between populations using VCFtools v.0.1.14[58] across 50 kb windows. Manhattan plots were generated using the Manhattan function in the qqman package in R[59]. Before plotting windowed $F_{ST}$ estimates, we filtered out all scaffolds with fewer than four windows and less than 10 SNPs. We also used VCFtools to calculate inbreeding coefficients ($F_{IS}$) for all individuals. We then calculated Tajima's D, nucleotide diversity (π), and absolute genetic divergence (Dxy) using the popgenWindows.py script (S. Martin; https://github.com/simonhmartin/genomics_general) to further provide insight into the evolutionary processes that have shaped population-level genetic variation. Putative chromosomal locations of different scaffolds were obtained by aligning them to the zebra finch (*Taeniopygia guttata*) assembly (bTaeGut1_v1.p; https://www.ncbi.nlm.nih.gov/datasets/genome/GCA_003957565.1/; SAMN02981239) using the pseudo-chromosome scaffolding command in SatsumaSynteny[60]. To analyze patterns of genetic structure among the four subspecies, Admixture

v.1.23[61] analyses were run using a filtered dataset (1,989,848 SNPs) that contained no missing data and was pruned to avoid linkage using the script ldPruning.sh (https://github.com/speciationgenomics/scripts/blob/master/ldPruning.sh). We investigated one to four population clusters with 200 bootstrap resampling iterations.

## Genome-wide divergence

We identified regions of elevated divergence among the four pairwise (or between size) comparisons including large- and smaller-bodied northern populations using the sliding window $F_{ST}$ estimates. Elevated values of $F_{ST}$ averaged over non-overlapping 50 kb windows were considered elevated if exceeding the 99.9th percentile of genome-wide mean value (i.e., >0.498 when comparing *sanaka* and *merrilli*, > 0.493 for *sanaka* and *rufina*, 0.507 for *maxima* and *rufina*, and 0.518 for *maxima* and *merrilli*; Fig. 2). We compiled a list of genes within these outlier windows using Geneious V.11.1.5[62]. To characterize putative candidate genes, we used ontology information from the zebra finch Ensembl database[63] the annotated song sparrow reference genome[50], and functional information from the Uniprot database[64] (Supplementary Table 2). We additionally compared the identified list of genes to known genes involved in recent analyses of body size or BMI in other species (see discussion; NHGRI-EBI GWAS catalog; https://www.ebi.ac.uk/gwas/).

To control for phenotypic differences between subspecies in traits that are not body size (e.g., plumage differences, migratory behavior), we compared the two smaller-bodied populations, *M. m. merrilli* and *M. m. rufina*, that also differ in migratory behavior as a measure of ensuring that the candidate genes are correlated with our trait of interest. We also compared the two large-bodied populations, *M. m. sanaka* and *M. m. maxima* as an additional control for our smaller- vs large-bodied comparisons. There were no overlaps in elevated peaks between our large-small bodied (between size) comparisons with those identified within our smaller-bodied or large-bodied (within size) control comparisons, increasing our confidence that we have identified genes related to body size variation as opposed to migratory behavior or other traits differing between subspecies.

Among our large- and smaller-bodied (between size) comparisons, we classified outlier regions as shared if two or more paired comparisons identified the same gene within 50 kb of an elevated window (Supplementary Table 2). We narrowed the candidate gene set by keeping only genes identified in all four comparisons between large- and smaller-bodied subspecies. Using this approach, we identified 9 genes that are shared in all of the between body size comparisons. Within these 9 genes, there are 467 variants across the four subspecies of interest.

## Signatures of selection

We used the candidate gene set, containing the 467 SNPs, to test for evidence of selective sweeps, identify SNPs highly correlated with body mass, and calculate the percentage of fixed or nearly fixed SNPs ($F_{ST} > 0.95$; Supplementary Table 5). To identify regions under selection, we calculated the composite likelihood ratio (CLR) test statistic using the program SweeD v3.3.2[65]. We determined the genotypes of all individuals by phasing and imputing missing data using Beagle v.3.3.2[66]. We ran SweeD separately for each population and on individual contigs of interest using the default parameters except for using a window size of 200 bp. The window within each contig with the highest CLR statistic is considered the likely location of a selective sweep[67].

## Hypothesis testing for validation

To test the hypothesis that our narrowed set of 9 candidate genes is related to body size, we predicted the genotypes of more distantly related small-bodied subspecies (*M. m. gouldii, M. m. heermanni, M. m.*

*samuelis, M. m. pusillula*, and *M. m. maxillaris*) from a different geographic region, the San Francisco Bay area of California. Whole-genome sequences for the 39 San Francisco Bay individuals were generated in a separate study[25] using library preparation and sequencing methods as detailed above. For the whole-genome sequencing summary statistics of the 39 San Francisco Bay individuals see Mikles and colleagues[25].

We first performed a partial Mantel test on the 3 contigs containing the 9 candidate genes across all 9 subspecies to assess whether patterns of divergence were due to neutral, including isolation by distance, or selective processes. Using the vegan[68] R package, we tested for correlations between mass and genetic distance, while accounting for geographic (Euclidean) distance.

To see how strongly allele frequency correlated with body size across all 79 individuals, we produced a matrix of the frequency of the non-reference allele using vcftools (--freq2 recode option) for the 467 SNPs. We used Pearson correlation coefficient ($r$) to quantify the relationship between the non-reference allele frequency and body mass (g), and between non-reference allele frequency and average summer and winter temperature (°C) at each of the SNPs. We selected the top 8 SNPs with a $r > 0.90$ between non-reference allele frequency and mass: contig 391 (chr 17) positions: 19152, 57980, 61835, 86656, 122228; contig 3361 (chr 2) positions: 255216, 255218, 262569. We also performed linear regression analyses to further explore the relationships between allele frequency and mass and temperature at each of the focal SNPs. For each analysis, allele frequency served as the independent variable, while mass and temperature were the dependent variables. Model parameters, including regression coefficients, intercepts, confidence intervals, effect sizes, R-squared values, p-values, and leverage and influence statistics can be found in Supplementary Tables 3 and 4. Relationships were then visualized using ggplot2[69] (Fig. 4).

In addition to looking at correlation between subspecies-level allele frequencies and mass, we assessed correlation with phenotype and genotype for individuals (heterozygotes or homozygotes for the reference/alternate alleles at our focal SNPs). To do this, we used the R package vegan[68] to calculate a distance matrix between the 8 SNPs identified above and included 1–3 additional SNPs immediately up and downstream of each focal SNP for visualization purposes (54 total SNPs). Less than three SNPs were used in cases where the SNP was located at the end of a contig or adjacent to another focal SNP. We then used the Ward algorithm as a grouping method to express the relationships between sites, which was implemented using the hclust function in vegan. We plotted the output using the plot.phylo function in the R package ape[70] and visualized genotypes by constructing a modified heatmap in which the base pairs of each individual at a locus are represented in the two halves of a diagonally split pixel (Supplementary Fig. 7).

## Reporting summary

Further information on research design is available in the Nature Portfolio Reporting Summary linked to this article.

# Data availability

The raw sequencing data generated in this study have been deposited in the National Center for Biotechnology Information (NCBI) BioProject database under accession code PRJNA1013697. Raw sequencing data for the California song sparrow subspecies used in this study are available in the NCBI BioProject database under accession code PRJNA1018990. Source data are provided with this paper.

# Code availability

Scripts and bioinformatic pipelines used in this study are available at https://github.com/kcarbeck/SOSP_body_size (https://doi.org/10.5281/zenodo.8365146).

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

## Acknowledgements

The authors thank the Cornell Lab of Ornithology, University of Alaska Museum, Berkeley Museum of Vertebrate Zoology, and Yvonne Chan for samples. B. Butcher, L. Campagna, D. Toews, T. Wang, and M. Whitlock provided essential assistance, comments, and support. Project funding was provided by the Natural Sciences and Engineering Research Council of Canada Discovery Grant RGPIN-2020-05268 (P.A.), University of British Columbia, and the Hesse Fellowship and Research Award (K.C.). The material in this manuscript is also based upon work supported by the National Science Foundation Postdoctoral Research Fellowship in Biology under grant No. DBI 1523719 (J.W.) as well as funding from the Fuller Evolutionary Biology Lab at the Cornell Lab of Ornithology (I.L.).

## Author contributions

K.C., P.A., and J.W. conceived and designed the study with input from I.L., K.W., and C.P. C.P. and K.W. conducted field work and collected samples. J.W. conducted laboratory work and K.C. carried out all bioinformatic analyses. Data analysis and interpretation were conducted by K.C. and J.W. with input from all co-authors. K.C. wrote the manuscript with input from all co-authors.

## Competing interests

The authors declare no competing interests.
