## [Peer Review File · Nature Communications]

REVIEWER COMMENTS

Reviewer #1 (Remarks to the Author):

This is an interesting paper that addresses a longstanding hypothesis in evolutionary biology: that organisms at higher latitudes evolve larger body sizes. By comparing two large and two small-bodied subspecies of song sparrows living at different latitudes, the authors identify nine candidate genomic regions exhibiting signatures of selection in all comparisons of small vs. large bodied subspecies. They validate the hypothesis that these loci are related to body size by using them to predict allele frequencies at nine SNPs in the coding regions of four genes in several small-bodied subspecies in California. This is a clever approach that goes a step beyond a standard outlier/ gene ontology analysis.

A primary challenge of this study though is that there is spatial autocorrelation between the trait of interest (body size) and both genetic and geographic distance between populations. This makes it difficult to convincingly determine whether candidate loci under selection are indeed associated with body size, or if allele frequencies differ between populations as a result of neutral processes or other unrelated forms of selection. In a standard GWAS you can control for geographic distance and genetic relatedness when detecting associations between traits and loci, but that kind of analysis was not used here (I assume because sample size is small). I do find the analyses the authors have done generally convincing given their sample size constraints, but I think they need to explicitly address the possibility of spatial autocorrelation and clearly explain why they are confident they are identifying signatures of selection related to body size and not allele frequency differences caused by other processes, e.g. isolation by distance or genetic drift in island populations. It might be worth exploring variance partitioning as a way to control for geographic & genetic distance when explaining phenotypic variation. For example, you might predict phenotypic distance between subspecies to be better explained by genetic distance at your candidate loci than by average genome-wide genetic distance and geographic distance in a multivariate analysis.

Paragraph starting line 88: A bit hard to follow in here. Does this mean that there were 25 elevated windows in each pairwise comparison (so 100 windows total)? And then there were nine genes that occurred in all large-small pairwise comparisons? It seems like you just focused on outlier regions that contained annotated genes- were there shared outlier regions in the large-small pairwise comparisons that didn't have annotated genes?

Paragraph starting line 99: this paragraph is hard to understand. I suggest summarizing numbers of pairwise fixed snps in a table (you could do this above the diagonal in the F_{st} supplemental table). I'm not sure how the islands factor in here. It seems like the key point is that if the fixed snps reflect fixation of alternative alleles associated with different body sizes, you should see more fixed snps in the large-small comparisons than large-large or small-small comparisons. It's hard to tell if that's the case as written.

Line 105: a bit odd to refer to estimates of π /Tajima's D as "evidence of past selection." How far in the past do you mean? Estimates of linkage disequilibrium/ recombination rate around loci under selection could give some estimate of how old selection is.

Line 119: I'm confused about how we've gone from 467 snps in 9 candidate genes shared between all pairwise large-small comparisons (lines 101/ 341/354) to 9 snps total (lines 123/356, Fig 4). How were these nine snps chosen from the 467 snps used to construct the genotype matrix? The lack of detail here makes the choice of snps in Figure 4 seem arbitrary (or like cherry picking), which undermines confidence in the results.

Line 138: As above- how did you choose the 54 snps to use in this phylogeny?

Line 144: also confusing- did you characterize 9 candidate genes or 9 candidate snps? Or both? Figure 4 only shows 9 snps in the coding regions of 4 genes.

Sentence starting line 151: seems like some words are missing here

Line 169: This sentence is awkwardly phrased

Line 224: Not sure why you reach this conclusion based on your data? If populations are mostly fixed for alternate alleles (Fig 4) it should diminish adaptive capacity. Is the idea that gene flow between populations would enable adaptation?

Line 228: I don't really understand this sentence. How have you ruled out lineage-specific selection in this study (e.g. a scenario where small body size is ancestral and large body size evolved in the lineage that expanded north)? In general, it would be helpful to see a phylogeny/ pairwise f_{st} / PCA between all the lineages included in this study (the 4 focal subspecies + the 5 validation subspecies). Are the small subspecies all more closely related to each other than to the large subspecies? What are the origins of the large bodied subspecies? A little bit of bio/phylogeographic context would help better support your conclusions.

Method question: Were invariant sites included in analyses of π and d_{xy} ? Use of only variant sites has been shown to produce downward biased estimates of π and d_{xy} (Korunes and Samuk 2021).

Reviewer #2 (Remarks to the Author):

This is a straight forward and interesting study that uses a nice system (song sparrows) and mainly genome resequencing data to investigate genetic mechanisms underlying Bergmann's Rule. The song sparrows are a great system for this purpose because it's one of most polytypic bird species in North America, with subspecies/populations spanning a wide range of ecogeographic environments. The study focused on 2 groups of birds: 2 large-bodied subspecies in the Aleutian Islands and 2 medium-sized subspecies in the west coast of Canada. Genome scans indicating strong, localized peaks implicate that genetic variation in 9 genes associate strongly with size differences. The authors further validated these results in 5 small-bodied subspecies from California. As supplemental supports, the authors surveyed (mainly human?) literature to show that some of these genes have been associated with body size and associated traits (eg BMI).

While the results are suggestive, it is my opinion that the authors need to carry out more analyses and studies to rule out other possibilities in order to tighten the conclusions. Members of each category of birds (ie large-, medium- and small-bodies birds) have ranges that are located near one another. Which means that they likely share similar histories, thus making each one a non-independent sampling point. For example, it is obvious that *maxima* and *sanaka* are likely part of the same stepwise population expansion from mainland Alaska. Under a population expansion scenario, it is possible some alleles can surf on the wave of population expansion and reach high frequency without providing any fitness benefits (L. Excoffier, M. Foll, R. m. J. Petit, *Annu. Rev. Ecol. Syst.*, (2008)). The lack of independent lineages (eg large-bodied birds from different parts of North America) means that the conclusion that "L227 genotypes of smaller-bodied subspecies are a consequence of historic selection acting locally on genetic variation shared among focal populations, rather than resulting from lineage-specific selection across ecological gradients" is trivial. Because birds from each size group are located close to each other, it is highly unlikely there is lineage-specific selection because each group shares the same history or could experience gene flow, allowing favored alleles to be selected for in all populations.

The authors also did not fully tease out whether high F_{st} values are the results of processes other than divergent selection. For example, the F_{st} outliers have low nucleotide diversity (see L107). Because F_{st} is a relative measure, low nucleotide diversity can cause a window to be an extreme outlier even if absolute divergence is not high (T. E. Cruickshank, M. W. Hahn, *Mol Ecol* 23, 3133 (2014)). This concern is partially mediated by D_{xy} comparisons. However, since candidate genes are selected based on F_{st} outliers, it is unclear whether a better exploration of the data will yield a different set (and how different?) of candidate genes. Averages of Tajima's D within outlier

windows were also compared against genome wide averages, but it isn't clear exactly what do the results mean. For example, merrilli (medium sized bird) has the most negative D, compared to sanaka (positive D) and maxima. Does this mean that these outlier windows have experienced the strongest selective sweep in merrilli? The authors also did not explore whether purifying selection in regions with low recombination rates played a role in shaping these outlier windows.

Additionally, while the groups of birds are clearly associated with size differences, and size differences with environmental/climatic differences, I feel that the current dataset cannot tease apart whether genetic variation explains size differences or, say, differences in thermoregulatory functions. Since the analyses focused on genes, descriptions of where the important SNPs fall (introns, exons, UTRs) and whether some may cause nonsynonymous changes will be useful too. Although it is possible/likely that these SNPs inside genes are not causal. Instead, causal SNPs may be found in regions containing cis-regulatory elements (but SNPs outside of genes were ignored).

Minor comments

Introduction

L41 This could be nip picky. Why should trait variation be affected by "past selection" and not by ongoing selection as well?

L62, Fig. 1. Since Bergmann's rule has 2 elements: a larger body mass helps heat retention during cold winters, and a smaller mass helps heat dissipation during hot summers, it will be interesting to split mean winter and summer temperatures. If two places have similar winter temperatures and different summer temperatures, it may show that the latter is the more important factor in driving body size differences.

Results

L80, Admixture results (Fig S1). Please show cross-validation error rate or likelihood support for K=2-4 models in Fig S1. It is unclear which subspecies (between maxima and sanaka) one individual belongs to.

L81 Please indicate whether the confidence intervals relate to SD or standard error of the mean.

L84, Fig S8. What do the hashed lines in Fig. S8 represent? Are they the 99.9th percentile for Dxy values? If so, how do they match up to Fig. 2? Is it possible that Dxy values in a few of the Fst outliers (in contig 391, towards the right of the figure) have lower than average values?

L100 Should it be "fixed and nearly fixed SNPs" as mentioned elsewhere in the paper (eg L342 in Methods)?

L101 The breakdown and description of different types of SNPs is confusing. There are a total of 467 SNPs within the 9 candidate genes. 25% of them are shared between 2 pairwise comparisons, but the next phrase pivots and discusses fixed SNPs in the other 2 pairwise comparisons. It appears to be a comparison of apples and oranges, and it is unclear also how this reflects history of the island subspecies.

L123 "These 9 SNPs included...". It is unclear how these 9 SNPs are selected. L356-358 in Methods also did not describe how these 9 SNPs were chosen out of all (467) that were present the 9 candidate genes.

L138 The 54 SNPs here came out of nowhere, and their descriptions in Methods are confusing too (see comments below).

Discussion

L192 If SNPs in 5 candidate genes identified with Alaska-Canada comparisons do not show up as highly differentiated in California birds, is it possible that the latter have not used variation in these genes as a way to achieve small size because these mutations have not arisen in California birds (but had arisen in the medium sized birds)? Perhaps a way to determine this is to find out if

alleles in these 5 genes in merilli and rufina are derived or ancestral.

Methods

L245 Merrilli was collected in Hyder, Alaska but Fig 1 does not show the range extending to Alaska.

L283 A minimum of 2X is too low. Heterozygotes are unlikely to be called accurately given this coverage.

L356 For the sake of consistency, SNPs should be referred as SNPs rather than loci. It is not clear why and how these 9 focal SNPs were chosen.

L359 The way it is written (plus minus ca. three loci), it is not obvious that it means that 1-3 SNPs up and downstream of each focal were chosen for distance matrix calculation. And why 3 SNPs? Further, the positions of these SNPs should be given to provide an indication of the potential for physical linkage with the focal SNPs. Why is hierarchical clustering used to estimate relationships among the combined genotypes, and not phylogenetic methods or SNP-based coalescent methods (eg SNAPP)?

RESPONSE TO REVIEWERS' COMMENTS

Comments from Reviewer #1

This is an interesting paper that addresses a longstanding hypothesis in evolutionary biology: that organisms at higher latitudes evolve larger body sizes. By comparing two large and two small-bodied subspecies of song sparrows living at different latitudes, the authors identify nine candidate genomic regions exhibiting signatures of selection in all comparisons of small vs. large-bodied subspecies. They validate the hypothesis that these loci are related to body size by using them to predict allele frequencies at nine SNPs in the coding regions of four genes in several small-bodied subspecies in California. This is a clever approach that goes a step beyond a standard outlier/ gene ontology analysis.

A primary challenge of this study though is that there is spatial autocorrelation between the trait of interest (body size) and both genetic and geographic distance between populations. This makes it difficult to convincingly determine whether candidate loci under selection are indeed associated with body size, or if allele frequencies differ between populations as a result of neutral processes or other unrelated forms of selection. In a standard GWAS you can control for geographic distance and genetic relatedness when detecting associations between traits and loci, but that kind of analysis was not used here (I assume because sample size is small). I do find the analyses the authors have done generally convincing given their sample size constraints, but I think they need to explicitly address the possibility of spatial autocorrelation and clearly explain why they are confident they are identifying signatures of selection related to body size and not allele frequency differences caused by other processes, e.g. isolation by distance or genetic drift in island populations. It might be worth exploring variance partitioning as a way to control for geographic & genetic distance when explaining phenotypic variation. For example, you might predict phenotypic distance between subspecies to be better explained by genetic distance at your candidate loci than by average genome-wide genetic distance and geographic distance in a multivariate analysis.

Thank you for your kind and helpful comments!

We agree and have worked to clarify this issue. While a GWAS might have solved some of issues regarding correlations between population structure and genotype-phenotype associations, the reviewer is correct that sample sizes for populations prohibited that type of analysis. There is a clear role for IBD and demography in structuring some of our populations, however we addressed the reviewers' concerns by first running a partial Mantel test and found a significant correlation between phenotype and genotype after controlling for geographic distance ($p = 0.004$; lines 83-86). Unfortunately, fully disentangling many of these processes is challenging as distance, latitude, and temperature are all correlated as well.

To address the reviewer's concerns regarding identifying signatures of selection, we have added an analysis to test for selective sweeps in the regions of our candidate genes (SweeD; Pavlidis et al. 2013). SweeD returned results confirming signals of selective sweeps associated with our

candidate genes, further suggesting selective mechanisms are responsible for driving body size variation in our populations (lines 111-113; Figure 3).

Paragraph starting line 88: A bit hard to follow in here. Does this mean that there were 25 elevated windows in each pairwise comparison (so 100 windows total)? And then there were nine genes that occurred in all large-small pairwise comparisons? It seems like you just focused on outlier regions that contained annotated genes- were there shared outlier regions in the large-small pairwise comparisons that didn't have annotated genes?

Thank you for pointing this out. We agree that the distinction between windows vs genes vs SNPs was confusing. We now define these terms at first mention and in the Methods and have revised the text to clarify all subsequent uses (lines 90-91; 97-99; and 343-349). Briefly, we identified, in total, 25 elevated windows across the 4 large- vs smaller-bodied subspecies comparisons. We chose to focus on genes only if they were shared between *all* 4 comparisons. Shared genes are defined as genes within 50 kb of the 25 elevated windows that are shared in all of our pairwise comparisons. This resulted in 9 shared genes containing 467 SNPs that we then examined closely, including in the SF Bay birds. We hope that our changes within the manuscript make all of this fully clear.

Paragraph starting line 99: this paragraph is hard to understand. I suggest summarizing numbers of pairwise fixed snps in a table (you could do this above the diagonal in the Fst supplemental table). I'm not sure how the islands factor in here. It seems like the key point is that if the fixed snps reflect fixation of alternative alleles associated with different body sizes, you should see more fixed snps in the large-small comparisons than large-large or small-small comparisons. It's hard to tell if that's the case as written.

In re-reading this paragraph, we agree that reporting numbers in a table rather than in text would be more suitable. We have largely moved this information to a supporting table (Supplementary Table 3) and condensed and combined this section with the previous paragraph (see lines 141-146).

Line 105: a bit odd to refer to estimates of π /Tajima's D as "evidence of past selection." How far in the past do you mean? Estimates of linkage disequilibrium/ recombination rate around loci under selection could give some estimate of how old selection is.

Thank you; we acknowledge the potential confusion implied. We have revised the text to read: "*We characterized genetic diversity and demographic history in all four northern subspecies by measuring nucleotide diversity (π) and Tajima's D across 50 kb windows.*"

Line 119: I'm confused about how we've gone from 467 snps in 9 candidate genes shared between all pairwise large-small comparisons (lines 101/ 341/354) to 9 snps total (lines 123/356, Fig 4). How were these nine snps chosen from the 467 snps used to construct the genotype matrix? The lack of detail here makes the choice of snps in Figure 4 seem arbitrary (or like cherry picking), which undermines confidence in the results.

We agree that this was confusing and appreciate this comment. We used correlation coefficients to quantify the strength of relationship between minor allele frequency (MAF) and body mass for all

SNPs on the 9 candidate genes noted (i.e., those identified as being shared in 4 pairwise comparisons of the Northern subspecies, see above response). We selected top outlier SNPs for those that had an $r^2 > 0.9$ between MAF and mass. This information has been added to both the methods (lines 368-375) and results (lines 134-136).

Line 138: As above- how did you choose the 54 snps to use in this phylogeny?

Based on the focal SNPs (see above), we chose 3 SNPs immediately up and downstream (only for visualization purposes). Given some coverage issues with the SF Bay birds as raised by reviewer two, we have largely downplayed this analysis and focus more on the correlations between MAF and mass at the focal SNPs. We still include the methods/results but with a caveat about issues with calling heterozygotes with lower coverage data (lines 152-156) and we have moved the old figure 4 to the supplement. Because we have higher coverage data with the northern subspecies, however, we have chosen not to remove the analyses completely but have worked to clarify our choice of SNPs for the figure (lines 379-381).

Line 144: also confusing- did you characterize 9 candidate genes or 9 candidate snps? Or both? Figure 4 only shows 9 snps in the coding regions of 4 genes.

See clarification to previous comments regarding genes versus SNPs.

Sentence starting line 151: seems like some words are missing here

Thank you for noting this issue. We have edited the sentence (line 165-168) to now read: *"Our observations of fixed and shared genomic differences, their co-variation with climate and body mass, and evident signatures of selection support Bergmann's Rule as influential in song sparrows, and reflect a capacity for and history of local adaptation to environment in this species."*

Line 169: This sentence is awkwardly phrased

Thank you; we agree the sentence distracted from the main point of the paragraph and have removed it.

Line 224: Not sure why you reach this conclusion based on your data? If populations are mostly fixed for alternate alleles (Fig 4) it should diminish adaptive capacity. Is the idea that gene flow between populations would enable adaptation?

We agree that this sentence was confusing. We have modified this paragraph, including this line (now line 228), to make more explicit connections between our conclusions and our results.

Line 228: I don't really understand this sentence. How have you ruled out lineage- specific selection in this study (e.g. a scenario where small body size is ancestral and large body size evolved in the lineage that expanded north)? In general, it would be helpful to see a phylogeny/ pairwise fst/ PCA between all the lineages included in this study (the 4 focal subspecies + the 5 validation subspecies). Are the small subspecies all more closely related to each other than to the large subspecies? What are the origins of the large bodied subspecies? A little bit of bio/phylogeographic context would help better support your conclusions.

Thank you for pointing this out. We have spent some time revising this paragraph to present our results more clearly. We now state:

“Our results reveal strong evidence of natural selection acting on genes which appear to contribute to phenotypic variation in body mass among nine subspecies distributed over a dramatic environmental cline, and they augment our prior work demonstrating local adaptation in the genomes of individuals across macro- to micro-geographic scales. Furthermore, our approach to validating gene function strongly suggests that the same genes have experienced historic and/or ongoing selection in multiple populations across the species’ range.”

We have also recreated the PCA to include all 9 subspecies, which shows that populations cluster by body size along the PC1 axis (12.5% of the variation explained). This has been included as a panel in figure 1.

Method question: Were invariant sites included in analyses of pi and dxy? Use of only variant sites has been shown to produce downward biased estimates of pi and dxy (Korunes and Samuk 2021).

Yes, invariant sites were included in our analyses of pi and dxy.

Comments from Reviewer #2

This is a straight forward and interesting study that uses a nice system (song sparrows) and mainly genome resequencing data to investigate genetic mechanisms underlying Bergmann’s Rule. The song sparrows are a great system for this purpose because it’s one of most polytypic bird species in North America, with subspecies/populations spanning a wide range of ecogeographic environments. The study focused on 2 groups of birds: 2 large-bodied subspecies in the Aleutian Islands and 2 medium-sized subspecies in the west coast of Canada. Genome scans indicating strong, localized peaks implicate that genetic variation in 9 genes associate strongly with size differences. The authors further validated these results in 5 small-bodied subspecies from California. As supplemental supports, the authors surveyed (mainly human?) literature to show that some of these genes have been associated with body size and associated traits (eg BMI).

Thank you for your kind words and feedback.

While the results are suggestive, it is my opinion that the authors need to carry out more analyses and studies to rule out other possibilities in order to tighten the conclusions. Members of each category of birds (ie large-, medium- and small-bodies birds) have ranges that are located near one another. Which means that they likely share similar histories, thus making each one a non-independent sampling point. For example, it is obvious that maxima and sanaka are likely part of the same stepwise population expansion from mainland Alaska. Under a population expansion scenario, it is possible some alleles can surf on the wave of population expansion and reach high frequency without providing any fitness

benefits (L. Excoffier, M. Foll, R. m. J. Petit, *Annu. Rev. Ecol. Syst.*, (2008)). The lack of independent lineages (eg large-bodied birds from different parts of North America) means that the conclusion that “L227 genotypes of smaller-bodied subspecies are a consequence of historic selection acting locally on genetic variation shared among focal populations, rather than resulting from lineage-specific selection across ecological gradients” is trivial. Because birds from each size group are located close to each other, it is highly unlikely there is lineage-specific selection because each group shares the same history or could experience gene flow, allowing favored alleles to be selected for in all populations.

We agree that additional analyses could be conducted to help quantify the direct versus correlated effects of natural selection on our identified candidate genes, and to test for the influence of processes such as random genetic drift. Reviewer 1 expressed similar thoughts. In contrast, we do not agree with Reviewer 2’s suggestion that it is necessary to reject all potential alternative explanations (especially those identified primarily via simulation; Excoffier et al. 2009) in order to suggest that our results provide strong empirical evidence of environmentally-induced selection on body size and the genes underlying it. We believe that our empirical predictions, comparative validation, and associated analyses of the candidate genes themselves together offer strong evidence in support of our main hypothesis and Bergmann’s Rule. However, because we also agree more can be done to explore the questions raised, we have tried in our revision to address those questions as best as we can with the data in hand.

Specifically, we first ran a partial Mantel test as a means of controlling statistically for the potential role of neutral forces linked to geographic isolation between individual genotype and phenotype. In support of our main hypothesis, we found a statistically significant relationship between genotype and body size after accounting for geographic distance, as expected if natural selection has influenced spatial variation in allele frequency. We have therefore added those results to the manuscript at lines 83-86 to substantially address reviewer concerns.

Second, we ran SweeD, a widely used method of detecting selective sweeps, on our candidate genes, which also returned results indicating strong support for the hypothesis that natural selection has acted on the candidate genes we associated with individual and population-level variation in body size (lines 111-113; composite likelihood ratio (CLR) peaks above 99th percentile of contig-wide mean). Because we do possess sufficient data to quantify the influence of phenotype or genotype on fitness directly, we offer the additional test above strengthen our suggestion that song sparrows display local adaptation in body mass.

We have also moderated our discussion of shared versus lineage specific selection (lines 228-234) and directly addressed the assumption that, in addition to these processes, neutral processes are likely to have played role in driving variation in the allele frequencies we report, and which are closely correlated to individual and population-level variation in body mass (lines 210-223).

Last, we suggest that better estimates of the relative contributions of the environment and geography on spatial patterns in genotype, while extremely useful, will require a fuller reconstruction of the demographic history of the 9 subspecies than we feel is necessary to support

our main points. Although not likely to be available for some months, we are attempting to conduct several related analyses to understand the roles of geography, environment, natural selection, and genetic drift in 21 of 25 subspecies of song sparrows described to date. We therefore hope to provide a more thorough assessment of these and related questions in the coming years. However, because divergence times across song sparrow populations are likely to be shallow, the computational requirements to do this work will require considerable effort and data. We therefore feel that a complete reconstruction the demographic history of this clade is beyond the scope of the current study.

The authors also did not fully tease out whether high F_{st} values are the results of processes other than divergent selection. For example, the F_{st} outliers have low nucleotide diversity (see L107). Because F_{st} is a relative measure, low nucleotide diversity can cause a window to be an extreme outlier even if absolute divergence is not high (T. E. Cruickshank, M. W. Hahn, *Mol Ecol* 23, 3133 (2014)). This concern is partially mediated by D_{xy} comparisons. However, since candidate genes are selected based on F_{st} outliers, it is unclear whether a better exploration of the data will yield a different set (and how different?) of candidate genes. Averages of Tajima's D within outlier windows were also compared against genome wide averages, but it isn't clear exactly what do the results mean. For example, *merrilli* (medium sized bird) has the most negative D , compared to *sanaka* (positive D) and *maxima*. Does this mean that these outlier windows have experienced the strongest selective sweep in *merrilli*? The authors also did not explore whether purifying selection in regions with low recombination rates played a role in shaping these outlier windows.

We agree that it is important to consider that low nucleotide diversity may increase the likelihood of F_{st} outliers even with relatively low absolute divergence. We note this limitation in our study (lines 126-127) and have taken steps to mitigate this concern by incorporating additional analyses, such as D_{xy} and SweeD, to help provide a more comprehensive understanding of the genetic differentiation and selective sweeps among populations. Our candidate genes were indeed selected based on F_{st} outliers, but this is the standard approach in this kind of comparison and although it comes with the caveats raised by the reviewer, we are not aware of alternative selection criteria against which we could test these outcomes.

Additionally, while the groups of birds are clearly associated with size differences, and size differences with environmental/climatic differences, I feel that the current dataset cannot tease apart whether genetic variation explains size differences or, say, differences in thermoregulatory functions. Since the analyses focused on genes, descriptions of where the important SNPs fall (introns, exons, UTRs) and whether some may cause nonsynonymous changes will be useful too. Although it is possible/likely that these SNPs inside genes are not causal. Instead, causal SNPs may be found in regions containing cis-regulatory elements (but SNPs outside of genes were ignored).

Thank you for the comment. We agree that the current dataset alone cannot definitively discern whether these SNPs are causative themselves. However, characterizing such variation allowed us to identify functional genes associated with individual and population-level variation in body mass, and further, to show variation within genes identified in 4 northern subspecies allowed us to predict body mass in a highly novel analysis of 5 California glades. Because clear regulatory

functions have been assigned to these genes in other species it seems quite likely to us that these SNPs have the potential to affect protein structure, function, or regulatory interactions. As such, our results are likely to compel further research into the molecular mechanisms underlying body size variation in vertebrates, whilst also providing strong support for Bergmann's Rule.

We also agree that providing a more detailed description of the location of the candidate SNPs would enhance the comprehensibility of our findings. Thus, we have modified line 138-141 in the results to clarify where SNPs fall within each candidate gene.

Regarding the consideration of SNPs outside of genes: we acknowledge that such regions may harbor cis-regulatory elements that simultaneously affect gene expression and influence the regulation in body size. We appreciate the reviewer's suggestion to investigate these SNPs further, but note that our goal was instead to identify potential candidate SNPs directly involved in body size regulation, rather than to describe and confirm in detail gene action and its regulatory nature. We have therefore not attempted to conduct additional analyses to fully characterize the regulatory nature or genetic architecture of the candidate genes or developed these potential points in the revised manuscript.

Minor comments

Introduction

L41 This could be a bit picky. Why should trait variation be affected by "past selection" and not by ongoing selection as well?

Agreed. This sentence has been modified accordingly.

L62, Fig. 1. Since Bergmann's rule has 2 elements: a larger body mass helps heat retention during cold winters, and a smaller mass helps heat dissipation during hot summers, it will be interesting to split mean winter and summer temperatures. If two places have similar winter temperatures and different summer temperatures, it may show that the latter is the more important factor in driving body size differences.

We agree that exploring winter and summer temperatures separately could provide valuable insights into the relative importance of heat/cold in driving body size differences. However, it is difficult to disentangle the specific effects of summer and winter temperatures. Each individual bird experiences both temperature regimes throughout the year, making it challenging to isolate their distinct effects on body size. Attempting to attribute body size differences solely to either summer or winter temperatures would introduce confounding factors and likely oversimplify the complex relationship between climate and phenotype. Additionally, other environmental factors, such as resource availability and predation pressure, may also vary across seasons and influence body size, which would require a complicated analysis that is beyond the scope of this paper and probably outside the explanatory power of the available data.

Results

L80, Admixture results (Fig S1). Please show cross-validation error rate or likelihood support for K=2-4 models in Fig S1. It is unclear which subspecies (between maxima and sanaka) one individual belongs to.

Thank you for the comment. We have added the following text to the figure legend of Fig S1 to support our interpretation: "...cross validation rates for K=2-4: 0.579; 0.605; 0.639, respectively".

L81 Please indicate whether the confidence intervals relate to SD or standard error of the mean.

Confidence intervals relate to SD. We have added this information to the text.

L84, Fig S8. What do the hashed lines in Fig. S8 represent? Are they the 99.9th percentile for Dxy values? If so, how do they match up to Fig. 2? Is it possible that Dxy values in a few of the Fst outliers (in contig 391, towards the right of the figure) have lower than average values?

Thank you for bringing this error in Figure S8 (now Supplementary Figure 6) to our attention. We mistakenly included the plot for *rufina-merrilli* that was labeled as the comparison of *sanaka-merrilli*. We apologize for any confusion this may have caused. We have updated the plot in the supplement to accurately represent the data for all pairwise comparisons (now including 2 control comparisons), and the hashed lines depicting the 99.9th percentile are noted in the figure legend.

L100 Should it be "fixed and nearly fixed SNPs" as mentioned elsewhere in the paper (eg L342 in Methods)?

We have modified this section to clarify how SNPs were chosen throughout. This section on fixed SNPs has also been changed.

L101 The breakdown and description of different types of SNPs is confusing. There are a total of 467 SNPs within the 9 candidate genes. 25% of them are shared between 2 pairwise comparisons, but the next phrase pivots and discusses fixed SNPs in the other 2 pairwise comparisons. It appears to be a comparison of apples and oranges, and it is unclear also how this reflects history of the island subspecies.

Agreed. Reviewer 1 had similar comments. We have worked to clarify this point throughout (please see response to reviewer 1 above).

L123 "These 9 SNPs included...". It is unclear how these 9 SNPs are selected. L356-358 in Methods also did not describe how these 9 SNPs were chosen out of all (467) that were present the 9 candidate genes.

Same as above. This was unclear, and we apologize for the confusion. We have added information to both the methods and results to address this (see response to reviewer 1 above).

L138 The 54 SNPs here came out of nowhere, and their descriptions in Methods are confusing too (see comments below).

Clarified throughout. See comment above.

Discussion

L192 If SNPs in 5 candidate genes identified with Alaska-Canada comparisons do not show up as highly differentiated in California birds, is it possible that the latter have not used variation in these genes as a way to achieve small size because these mutations have not arisen in California birds (but had arisen in the medium sized birds)? Perhaps a way to determine this is to find out if alleles in these 5 genes in *merrilli* and *rufina* are derived or ancestral.

We appreciate this interesting suggestion and will consider how/whether such analyses might be conducted in future. However, we expect that effort expended to explore such patterns will be better rewarded with continent-wide analyses now underway in hand. As Reviewer 1 notes above, having multiple opportunities to contrast patterns of genetic variation across similar environment clines and/or pairwise contrasted is likely to offer a more synthetic and satisfying outcome than our current data allow.

Methods

L245 *Merrilli* was collected in Hyder, Alaska but Fig 1 does not show the range extending to Alaska.

We have re-examined Figure 1, we can confirm that it does depict *merrilli*'s range extending into southern Alaska (including Hyder, AK). We therefore assume the reviewer missed seeing that when conducting their review.

L283 A minimum of 2X is too low. Heterozygotes are unlikely to be called accurately given this coverage.

This is a valid point. This cut-off was initially chosen for the SF Bay birds, which had lower coverage sequencing. To address this, we have redone our analyses using average allele frequencies for each subspecies (see lines 368-386 in the methods and lines 136-156 in the results). Our new results are highly consistent with our prior results, but now based on population averages. We have retained the individual-level genotype methods/results but moved the main figure to the supplement, and we have added the caveats noted regarding coverage. Given the higher coverage of the 4 northern subspecies, we feel that our results remain informative and relevant to our goal and were hesitant to remove them entirely. We hope that our effort to conduct additional analyses, note potential weakness in the CA reads, and allow readers to see the results of all analyses should they wish to do so is acceptable. We of course remain willing to consider other suggestions.

L356 For the sake of consistency, SNPs should be referred as SNPs rather than loci. It is not clear why and how these 9 focal SNPs were chosen.

We have accepted the reviewer's suggestion and changed the word "loci" in this sentence to "SNPs". Additionally, we have edited the text to elaborate how the 9 SNPs were selected (see lines 368-375).

L359 The way it is written (plus minus ca. three loci), it is not obvious that it means that 1-3 SNPs up and downstream of each focal were chosen for distance matrix calculation. And why 3 SNPs? Further, the positions of these SNPs should be given to provide an indication of the potential for physical linkage with the focal SNPs. Why is hierarchical clustering used to estimate relationships among the combined genotypes, and not phylogenetic methods or SNP-based coalescent methods (eg SNAPP)?

Thank you for pointing this out. We have modified this section to clarify why SNPs were chosen. However, given the comment above regarding coverage, we have largely re-worked this section to focus more on allele frequencies by subspecies and correlations with size and temperature. We think that this new approach is more robust. While we retain the distance matrices/clustering in the supplement, this section is significantly reduced and thus we feel that additional tree-based analyses are beyond the scope of our study.

REVIEWERS' COMMENTS

Reviewer #1 (Remarks to the Author):

The authors have done a commendable job revising their manuscript. The analyses of selective sweeps reinforce their conclusions and they have clarified areas of confusion. I have a few minor comments:

The two new paragraphs on evidence for selection in the Methods feel like they're in the wrong spot: they're in the middle of the section about validation of candidate genes in the small San Francisco subspecies, which is confusing. I suggest adding a separate section in the Methods called "Signatures of Selection" that comes before the validation section to mirror the order of the Results.

Line 147: phylogenetic

Figure 1 caption: some words missing in the sentence that starts "Inset map shows..."

Reviewer #2 (Remarks to the Author):

Major comments.

The authors did a nice job improving the manuscript and responding to reviewers' comments. I only have the following minor comments.

Minor comments

L38

Should be "historical".

L84-85.

I am trying to understand the meaning of a significant correlation between genetic distance and phenotype after correcting for distance. Because the genetic distances are genome-wide distances (and not based on outlier regions), this significant residual correlation should still mainly reflect neutral processes or shared ancestry, not natural selection. Is this a correct interpretation?

L89

I think it's more clear to say something like "25 elevated windows.. shared by at least 2 pairwise comparisons of large- and smaller-bodied subspecies.."

L92

"contained annotated genes.. which we identified as genes.."

The way it's written is somewhat clumsy.

L96

"... only a small proportion of the genome is ..."

Although this statement is not wrong, but I think the idea is obvious and should be expressed in a different way. By finding genes in 0.1% of the windows, one is obviously talking about a very small portion of the genome. I think the stronger evidence for supporting selection is that F_{st} values in outlier windows are so high against generally low genome-wide averages.

L112

I am assuming that evidence of selective sweep is referring to the results of SweeD. This should be explicitly related to the CLR plots of figure 3.

L114

Tajima's D values can be used to make inferences about demographic history (eg more negative values indicate expansion). However, based on the context, it is obviously used to determine if a

region has experienced a recent sweep. Therefore, I don't think you should say that demographic history is being characterized.

L134

Would the term "allele frequency of non-reference allele" be more appropriate? Since these alleles are the major alleles in the large bodied birds.

Fig 4

The significance of each correlation is obviously very much driven by allele frequencies of the 2 large-bodied populations. Since R^2 and standard error are given, I wonder if the authors had conducted a regression analysis, rather than a correlation analysis (and R^2 is coefficient of determination, not correlation coefficient, L370). In this case, in view of 2 data points being so important (ie the 2 large-bodied populations) in every correlation/regression, leverage and influence analyses should be carried out.

L147

It is not clear where the number 54 comes from. It should be mentioned at this point that additional 1-3 SNPs up and downstream of each focal SNP are being included (legend of Fig S7 mistakenly says "three SNPs surrounding the eight SNPs of interest.") And it should be mentioned why fewer than 3 were used at times (exceeding certain genomic distance? Focal SNP is located at the end of a contig?).

Fig. 3

I am curious if the authors found the other half of the chromosome 17 peak. It feels that there is another contig (adjacent to 391) with high F_{st} at its 5' end.

RESPONSE TO REVIEWERS' COMMENTS

Reviewer #1

The authors have done a commendable job revising their manuscript. The analyses of selective sweeps reinforce their conclusions and they have clarified areas of confusion. I have a few minor comments:

The two new paragraphs on evidence for selection in the Methods feel like they're in the wrong spot: they're in the middle of the section about validation of candidate genes in the small San Francisco subspecies, which is confusing. I suggest adding a separate section in the Methods called "Signatures of Selection" that comes before the validation section to mirror the order of the Results.

Thank you for your kind remarks and helpful comments.

We agree that the section of text about evidence of selection was in the wrong spot. We have taken the reviewer's suggestion and added a separate Methods section called "Signatures of Selection" that now comes before the validation section.

Line 147: phylogenetic

Thank you for pointing this out this error; we have made this edit.

Figure 1 caption: some words missing in the sentence that starts "Inset map shows..."

Thank you for noting this. We have updated this sentence to now read: "*Inset map shows the ranges of small-bodied San Francisco Bay subspecies (outlined in red) that were used in analyses to validate candidate genes.***"**

Reviewer #2

Major comments.

The authors did a nice job improving the manuscript and responding to reviewers' comments. I only have the following minor comments.

Thank you for your kind and helpful feedback.

Minor comments

L38

Should be "historical".

Thank you for pointing this out this error; we have edited accordingly.

L84-85.

I am trying to understand the meaning of a significant correlation between genetic distance and phenotype after correcting for distance. Because the genetic distances are genome-wide distances (and not based on outlier regions), this significant residual correlation should still mainly reflect neutral processes or shared ancestry, not natural selection. Is this a correct interpretation?

Thank you for this comment. After re-reading this sentence in light of this comment, we understand where the confusion is coming from. The text was previously located in a spot in the manuscript that didn't provide the proper context. We now include the Mantel test results on line 150 (below) and have elaborated in both the results and methods to clarify these results.

“We used a partial Mantel test to further explore whether these relationships reflect a history of selection versus neutral processes using the 3 contigs containing the 9 candidate genes. Pairwise genetic distance and phenotype were significantly correlated across all 9 subspecies after controlling for geographic distance (partial Mantel test: $r = 0.183$; $p = 0.004$).”

L89

I think it's more clear to say something like “25 elevated windows.. shared by at least 2 pairwise comparisons of large- and smaller-bodied subspecies..”

Thank you, we have taken this suggestion and the sentence now reads: “We identified a total of 25 elevated windows of F_{ST} that were shared by at least two pairwise comparisons of large- and smaller-bodied subspecies...”.

L92

“contained annotated genes.. which we identified as genes..”

The way it's written is somewhat clumsy.

We agree that this sentence was redundant. We have removed the second clause and it to now reads: “Several of these shared elevated 50 kb windows contained annotated genes (13 genes: sanaka and merrilli; 10: sanaka and rufina; 18: maxima and rufina; 19: maxima and merrilli).”

L96

“... only a small proportion of the genome is ...”

Although this statement is not wrong, but I think the idea is obvious and should be expressed in a different way. By finding genes in 0.1% of the windows, one is obviously talking about a very

small portion of the genome. I think the stronger evidence for supporting selection is that F_{ST} values in outlier windows are so high against generally low genome-wide averages.

Thank you, we appreciate this suggestion. We have removed this sentence and added the following text to highlight the high F_{ST} values against relatively low background genome-wide F_{ST} . This line now reads: *“Overall, the mean F_{ST} values in the 50 kb outlier windows were elevated for each of the four pairwise comparisons (Mean F_{ST} : sanaka-merrilli = 0.671; sanaka-rufina = 0.625; maxima-rufina = 0.645; maxima-merrilli = 0.828), relative to low genome-wide F_{ST} averages (sanaka-merrilli = 0.159; sanaka-rufina = 0.176; maxima-rufina = 0.205; maxima-merrilli = 0.247).”*

L112

I am assuming that evidence of selective sweep is referring to the results of SweeD. This should be explicitly related to the CLR plots of figure 3.

We have clarified on this line to now read: *“To search for evidence of local adaptation in body size, we calculated the composite likelihood ratio (CLR) test statistic to test for the presence of selective sweeps on the 3 contigs containing the 9 candidate genes (Fig. 3).”*

L114

Tajima's D values can be used to make inferences about demographic history (eg more negative values indicate expansion). However, based on the context, it is obviously used to determine if a region has experienced a recent sweep. Therefore, I don't think you should say that demographic history is being characterized.

Thank you for this suggestion. We have revised the wording in this sentence to reflect our use of Tajima's D to characterize selection. It now reads: *“We also measured nucleotide diversity (π) and Tajima's D across 50 kb windows in all four northern subspecies to characterize genetic diversity and detect evidence of selection.”*

L134

Would the term “allele frequency of non-reference allele” be more appropriate? Since these alleles are the major alleles in the large bodied birds.

We appreciate this suggestion and have changed the phrasing from minor allele frequency to either “allele frequency of the non-reference allele” or “non-reference allele frequency” throughout.

Fig 4

The significance of each correlation is obviously very much driven by allele frequencies of the 2 large-bodied populations. Since R^2 and standard error are given, I wonder if the authors had conducted a regression analysis, rather than a correlation analysis (and R^2 is coefficient of determination, not correlation coefficient, L370). In this case, in view of 2 data points being so important (ie the 2 large-bodied populations) in every correlation/regression, leverage and influence analyses should be carried out.

Thank you for this comment. Previously we had only conducted a correlation analysis, however upon the reviewer's suggestion we have added the results of a regression analysis, including a leverage and influence analysis (see Supplementary Tables 3 – 4). These changes are also reflected in the methods (lines 396 – 402):

“We also performed linear regression analyses to further explore the relationships between allele frequency and mass and temperature at each of the focal SNPs. For each analysis, allele frequency served as the independent variable, while mass and temperature were the dependent variables. Model parameters, including regression coefficients, intercepts, confidence intervals, effect sizes, R-squared values, p-values, and leverage and influence statistics can be found in Supplementary Tables 3 and 4.”

Results (lines 146 – 150):

“We observed consistent positive regressions of mass on non-reference allele frequency at each of the 8 focal SNPs (range: $F(1,7) = 31.96 - 142.66$; range $R^2: 0.820 - 0.953$; $p < 0.001$; Supplementary Table 3). In contrast, regressing temperature on allele frequency revealed negative relationships at each of the focal SNPs (range: $F(1,7) = 11.38 - 18.15$; range $R^2: 0.619 - 0.722$; $p < 0.05$; Supplementary Table 4).”

L147

It is not clear where the number 54 comes from. It should be mentioned at this point that additional 1-3 SNPs up and downstream of each focal SNP are being included (legend of Fig S7 mistakenly says “three SNPs surrounding the eight SNPs of interest.”) And it should be mentioned why fewer than 3 were used at times (exceeding certain genomic distance? Focal SNP is located at the end of a contig?).

We agree this sentence was ambiguous. We have edited the sentence to now read: “*The phylogenetic tree reconstructed from individual genotypes of the 8 focal SNPs and 1-3 additional SNPs immediately up and downstream (54 total SNPs; Supplementary Fig. 7; see Hypothesis Testing for validation section in Methods) was consistent with the strong correlation between body size and MAF across subspecies.*”

We have also edited the legend of Supplementary Figure 7 to clarify that 1-3 SNPs upstream and downstream were included.

Additionally, we clarified why less than three SNPs were used at times in the methods on line 408: *“Less than three SNPs were used in cases where the SNP was located at the end of a contig or adjacent to another focal SNP.”*

Fig. 3

I am curious if the authors found the other half of the chromosome 17 peak. It feels that there is another contig (adjacent to 391) with high Fst at its 5' end.

We were also curious about this. For this reason, we used satsuma to assemble our song sparrow genome to the zebra finch reference genome to get a chromosome level assembly. However, the peak on chromosome 17/contig391 is in fact at the end of the chromosome.